# Mixture of Demonstrations for In-Context Learning

**Song Wang**[*]
University of Virginia
sw3wv@virginia.edu

**Zihan Chen**[*]
University of Virginia
brf3rx@virginia.edu

**Chengshuai Shi**
University of Virginia
cs7ync@virginia.edu

**Cong Shen**
University of Virginia
cong@virginia.edu

**Jundong Li**
University of Virginia
jundong@virginia.edu

## Abstract

In-Context Learning (ICL) empowers Large Language Models (LLMs) to tackle various tasks by providing input-output examples as additional inputs, referred to as demonstrations. Nevertheless, the performance of ICL could be easily impacted by the quality of selected demonstrations. Existing efforts generally learn a retriever model to score each demonstration for selecting suitable demonstrations, however, the effect is suboptimal due to the large search space and the noise from unhelpful demonstrations. In this study, we introduce **MoD** (**M**ixture **o**f **D**emonstrations), which partitions the demonstration pool into groups, each governed by an expert to reduce search space. We further design an expert-wise training strategy to alleviate the impact of unhelpful demonstrations when optimizing the retriever model. During inference, experts collaboratively retrieve demonstrations for the input query to enhance the ICL performance. We validate MoD via experiments across a range of NLP datasets and tasks, demonstrating its state-of-the-art performance and shedding new light on the future design of retrieval methods for ICL.

## 1 Introduction

Large language models (LLMs) have demonstrated remarkable potential across various natural language processing (NLP) tasks [62, 43, 6], such as semantic parsing [22, 53] and commonsense reasoning [42, 61]. However, the large parameter size of these models often comes with significant costs for retraining or fine-tuning when they are applied to novel tasks [16, 25, 59]. Fortunately, as LLMs increase in size, they acquire the *In-Context Learning* (ICL) capability [50, 47], wherein the model can achieve significant performance improvements when provided with a limited number of demonstration examples during inference, without updating model parameters [5].

Although ICL has exhibited promising performance in various tasks, this capability also introduces a challenge related to robustness [5, 15, 36, 29]: ICL is highly sensitive to the selection of in-context demonstrations, and suboptimal selections could even lead to worse performance than random selections [34, 27, 26]. Recently, extensive research efforts have been dedicated to improving the selection of in-context demonstrations [47, 35]. For example, learning-free methods directly select demonstrations according to the similarity of demonstration embeddings from a pre-trained encoder [55]. Learning-based methods generally optimize a retriever based on feedback or supervision signals (e.g., output probabilities) from LLMs, and demonstrate superior performance compared to learning-free methods [34, 57].

However, the performance of these approaches is limited by two crucial challenges. (1) **Large Search Space.** As ICL requires the retrieval of multiple demonstrations from a sample pool, it is

---

[*]indicates equal contributions, random order.

38th Conference on Neural Information Processing Systems (NeurIPS 2024).

difficult to retrieve the optimal set of demonstrations from such a large search space, especially when the available sample pool is more extensive. Moreover, the total number of possible retrieval outcomes grows exponentially as the size of the retrieved set increases, rendering the retrieval even more challenging. (2) **Insufficient Optimization.** Existing learning-based works generally optimize the retriever model by preferring demonstrations that could aid model predictions. However, the common practice of randomly sampling a demonstration set in each training step could be suboptimal. For example, the samples in the entire set may contribute differently to or even impair the model predictions, but they are assigned the same retrieval scores, which could make the optimized model prefer the less helpful demonstrations.

To address the above challenges, we propose a novel demonstration retrieval framework named **MoD** (**M**ixture **o**f **D**emonstrations) that effectively navigates the sample pool while enabling precise optimization for beneficial demonstrations. First, to deal with the challenge of large search space, we leverage the mixture of experts (MoE) mechanism [18, 48] and partition the demonstration pool into distinct groups, each considered as an expert. Subsequently, we train an individual retriever model for each expert to prioritize helpful demonstrations, and during inference, we aggregate demonstrations retrieved from experts as the final demonstration set. Such a design largely reduces the search space of retrieval while also ensuring diversity in the demonstration set without sacrificing performance. Second, to tackle the problem of insufficient optimization, we propose a novel training strategy drawing inspiration from coordinate descent (CD) [54], which iteratively optimizes each dimension of a variable while fixing other dimensions. Inspired by CD, we propose an expert-wise training strategy that learns the retrieval score of any candidate demonstration while pairing it with demonstrations selected by all experts. These demonstrations are fixed while we only optimize one candidate demonstration at each step. As a result, we could ensure that all demonstrations used for optimization are optimal (except the candidate demonstration), thereby mitigating the disruption from unhelpful demonstrations. In summary, our contributions are as follows:

- We propose a novel demonstration retrieval framework MoD that learns multiple experts to collaboratively select demonstrations across the entire sample pool.

- Our design of multiple experts and expert-wise training could deal with the challenge of large search space and insufficient optimization, which have not been thoroughly investigated before.

- We conduct extensive experiments across a variety of NLP tasks to evaluate our framework in retrieving suitable demonstrations for ICL. The results demonstrate the superior performance of MoD over other state-of-the-art baselines.

## 2  Related Works

**In-Context Learning.**  In-context learning (ICL) empowers large language models (LMs) by providing them with a few input-output examples as demonstrations [5], enabling them to 'learn by analogy' and proficiently undertake intricate tasks, such as machine translation [1, 39], data generation [56], and others [49, 13, 30]. Although successful in many aspects, the efficacy of ICL is frequently hindered by its sensitivity to the selection of in-context examples, prompting research into optimized selection strategies [26, 27, 63]. These selection techniques can be classified into learning-free and learning-based methods. Learning-free methods typically employ heuristic criteria for selecting demonstrations without directly querying LLMs during the selection process. These criteria include assessing semantic similarity between testing examples and demonstrations [26], measuring entropy [27], and ensuring diversity [41, 21, 1]. However, these methods do not actively engage with LLMs and often result in suboptimal performance. In contrast, researchers leverage feedback from LLMs as supervision signals to explore more advanced learning-based methods. For instance, EPR [34] trains a singleton example scorer using contrastive learning with signals from LM inference. Furthermore, UDR [23] extends EPR in a unified formulation. These methods, however, do not account for interactions between in-context examples. In comparison, CEIL [57]tackles this challenge by jointly modeling the selection of the exemplar set and training a retriever to score the exemplar set. Nonetheless, CEIL faces challenges such as exponential search space in the size of the demonstration pool. To address this, it narrows down the candidate space using a $K$-NN retriever before the selection stage, potentially leading to suboptimal demonstration sets due to insufficient exploration of the entire demonstration pool.

**Mixture of Experts.** The idea behind Mixture of Experts (MoE) is to have a set of expert networks, each specializing in a particular task or a subset of the input space [38, 45, 19]. Wang et al. extended this paradigm to the prompt optimization task, achieving substantial performance improvements [48]. However, their approach overlooks the potential benefits of leveraging multiple expert collaborations. We extend the MoE framework to tackle the demonstration selection problem, aiming to effectively navigate the demonstration pool while considering the interplay among in-context examples.

## 3 Methodology

### 3.1 Problem Setup

Given a set $\mathcal{D} = \{e_i\}_{i=1}^n = \{(x_i, y_i)\}_{i=1}^n$ of input-output pairs (referred to as the demonstration pool), and a test example $(x_{test}, y_{test}) \in \mathcal{D}_{test}$, the strategy of ICL is to retrieve a set of demonstrations $\mathcal{S}(x_{test}) \in \{\mathcal{S} | \mathcal{S} \subseteq \mathcal{D}, |\mathcal{S}| = L\}$, which serves as the input conditioning for a pretrained LLM $\mathcal{M}$ to make predictions on $x_{test}$:

$$\hat{y} = \mathrm{argmax}_y \mathcal{P}_\mathcal{M}(y \,|\, \mathcal{S}(x_{test}), x_{test}). \tag{1}$$

where $\mathcal{P}_\mathcal{M}$ measures the likelihood of a candidate answer $y$ generated by $\mathcal{M}$. We aim to provide the proper demonstration set $\mathcal{S}(x_{test})$ for each $x_{test}$ that helps $\mathcal{M}$ make good predictions on $x_{test}$. However, the search space could be $|\mathcal{D}|^L$, which is computationally infeasible for an exhaustive search. To deal with this, existing works have proposed to learn an embedding for retrieval or narrow down the search space with a KNN retriever. Such strategies are suboptimal as they ignore demonstrations that are far from the input, in terms of embedding similarities. However, such demonstrations could still be useful for ICL [21, 41].

We introduce our proposed method as the **M**ixture **o**f **D**emonstrations (MoD) and outline its demonstration assignment, expert's retriever training, and inference as follows.

### 3.2 Mixture of Demonstration (MoD) Framework

To address the aforementioned challenges of an extremely large search space, we propose a novel mixture of demonstration (MoD) framework based on the mixture of experts (MoE) paradigm [18]. Specifically, we partition the demonstration pool into distinct groups, each governed by an expert. For each expert, we train a unique retriever, implemented as a scorer function, to select suitable demonstrations for the test example $x_{test}$. During the training of the experts' retrievers, we consider the interactions among demonstrations in the prompt. With our MoD framework, the demonstration selection process for ICL is transformed into an expert assignment problem along with an individual retrieval task for each of the assigned experts. The optimal retrieved set of demonstrations for $x_{test}$ could be achieved by selecting demonstrations from the most relevant experts, represented as follows:

$$\mathcal{S}(x_{test}) = \bigcup_{i=1}^C \mathrm{argmax}_{\widehat{\mathcal{S}}_i \subseteq \mathcal{C}_i} \sum_{e \in \widehat{\mathcal{S}}_i} g_i(x_{test}, e), \text{ where } |\widehat{\mathcal{S}}_i| = \lfloor h(\mathcal{C}_i, x_{test}) * L \rfloor, \text{and } \mathcal{D} = \bigcup_{i=1}^C \mathcal{C}_i. \tag{2}$$

Here $\mathcal{S}(x_{test})$ represents the set of demonstrations selected for the test example $x_{test}$. $C$ is the total number of experts into which the dataset $\mathcal{D}$ is divided, and $\mathcal{C}_i$ represents the distinct demonstration set of the $i$-th expert. $\widehat{\mathcal{S}}_i$ is the set of demonstrations selected from $\mathcal{C}_i$ while maximizing the sum of values given by the scorer function $g_i(\cdot)$ of the $i$-th expert, which measures the importance of the demonstration $e$ from $\mathcal{C}_i$ with respect to the test example $x_{test}$. $h(\mathcal{C}_i, x_{test})$ is a function that determines the relevance between $x_{test}$ and each expert $\mathcal{C}_i$ and also indicates the ratio of demonstrations from this expert in $\mathcal{S}(x_{test})$. With Eq. (2), we could select the most helpful demonstrations from relevant experts, regarding any input test sample $x_{test}$. Our demonstration selection strategy of using multiple experts could efficiently cover the entire search space without high computational costs, as specific experts will be omitted during retrieval when $\lfloor h(\mathcal{C}_i, x_{test}) * L \rfloor = 0$. Our strategy also enables the retrieval of dissimilar samples that could be helpful for ICL, as we cover multiple experts across the entire search space. In concrete, by optimizing the scorer function $g_i$ of each expert, we could retrieve the demonstration set $\mathcal{S}(x_{test})$ that could maximally aid in ICL for $x_{text}$. In the following, we introduce details of the two-step retrieval process in our framework: 1) Demonstration Assignment and 2) Expert Retrieval.

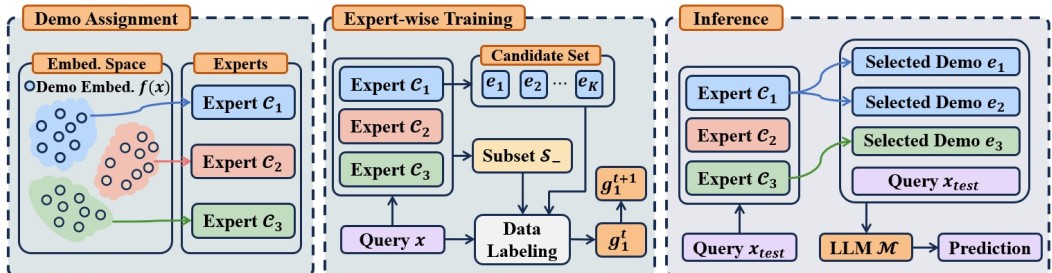

Figure 1: The overall process of our MoD framework. Before training, we first assign a set of demonstrations to each of the experts. Then we perform expert-wise training to obtain a retriever model for each of the experts. We ensure that the subset $\mathcal{S}_-$ is optimally selected from all experts to filter out unhelpful demonstrations during training. During inference, multiple experts will provide demonstrations for predictions on the input query.

## 3.3  Demonstration Assignment

We first introduce the strategy of partitioning the entire demonstration set and assigning the corresponding demonstrations to experts. Previous studies have demonstrated that selecting demonstrative samples $x_i$ with smaller distances between them and $x_{test}$ in the sentence embedding space can enhance the effectiveness of ICL [26, 41, 34]. Based on these findings, we propose to ensure that demonstrations assigned to a specific expert should be similar. Therefore, we employ the K-means clustering approach to partition the demonstration set $\mathcal{D} = \{e_i\}_{i=1}^n = \{(x_i, y_i)\}_{i=1}^n$ into $C$ clusters $\{\mathcal{C}_1, \mathcal{C}_2, ..., \mathcal{C}_C\}$ based on embedding distances, and demonstrations in each cluster are assigned to a specific expert. In this way, each cluster comprises semantically similar demonstrations, from which the corresponding expert selects suitable ones for $x_{test}$. Specifically, we utilize the widely-used Sentence-BERT model [32] as the embedding model $f(\cdot)$ [41, 34]. To adaptively obtain the optimal number of clusters $C$, we combine the within-cluster sum of squared errors with a regularization term to constrain $C$. The criterion can be expressed as follows:

$$C = \underset{C}{\operatorname{argmin}} \sum_{k=1}^{C} \sum_{(x_i, y_i) \in \mathcal{C}_k} \|f(x_i) - \mu_k\|^2 + \lambda C, \text{ where } \mu_k = \frac{1}{|\mathcal{C}_k|} \sum_{(x_i, y_i) \in \mathcal{C}_k} f(x_i). \quad (3)$$

Here, $\mathcal{C}_k$ is the $k$-th cluster, and $\mu_k$ denotes its centroid. With the obtained clusters, given an input test sample $x_{test}$, we compute its similarity to the centroid of any expert $i$ in the embedding space as follows:

$$h(\mathcal{C}_i, x_{test}) = \cos(f(x_{test}), \mu_i). \quad (4)$$

Here $f(x)$ is the learned embedding of sample $x$. With the obtained scores regarding each expert, we could determine the number of demonstrations selected from each expert as $|\widehat{\mathcal{S}_i}| = \lfloor h(\mathcal{C}_i, x_{test}) * L \rfloor$.

## 3.4  Expert-wise Training of Retriever Models

**Optimization Objective.**  In this subsection, we introduce our approach for training a demonstration retriever, implemented as a scorer function $g_i(\cdot)$, for each expert $i$. It is essential that the primary objective for the retriever is to select appropriate demonstrations based on the few-shot pattern in ICL. Therefore, considering the interaction among demonstrations, the search space can be as large as $|\mathcal{D}|^L$, where $L$ is the number of demonstrations used in ICL [57]. To mitigate the computational burden associated with such a large search space, we draw inspiration from the concept of coordinate descent (CD) [54]. CD optimizes a variable iteratively by fixing most dimensions of the variable vector at their current values and approximately minimizing the objective. In this manner, the optimization problem in each step has fewer dimensions, making the optimization easier compared to directly optimizing all dimensions. In concrete, we propose the following optimization objective for training the scorer function $g_i(\cdot)$ of expert $i$:

$$\phi_i^* = \underset{\phi}{\operatorname{argmax}} \mathbb{E}_{(x_{test}, y_{test}) \in \mathcal{D}_{test}} \mathcal{L}(y_{test}, x_{test}, \{e_{test}^{i_*}\} \cup \mathcal{S}_-(x_{test})), \quad (5)$$

where $\phi_i^*$ represents the optimal parameters of $g_i(\cdot)$. $\mathcal{L}$ is an evaluation criterion and can encompass various metrics, such as the log-probability of the output, i.e., $\mathcal{L}(y, x, \mathcal{S}) := \mathcal{P}_{\mathcal{M}}(y \mid \mathcal{S}, x)$, indicating

the utility of $\mathcal{S}$ for decoding the target answer [57]. $\mathcal{S}_-(x_{test})$ denotes the demonstration set retrieved based on Eq. (2), except that the value of $L$ in it is replaced with $L-1$. Additionally, $e_{test}^{i_*}$ represents the sample with the highest score in the unselected set from $\mathcal{C}_i$ with respect to the test example $x_{test}$, i.e.,

$$e_{test}^{i_*} = \underset{e \in \mathcal{C}_i \setminus \mathcal{S}_-(x_{test})}{\operatorname{argmax}} g_i(x_{test}, e). \tag{6}$$

In other words, akin to how CD optimizes one component while fixing others, our objective is to optimize $g_i$ such that we can retrieve the demonstration (i.e., $e_{test}^{i_*}$) that contributes the most to ICL when $L-1$ demonstrations (i.e., $\mathcal{S}_-(x_{test})$) are already retrieved and fixed. After iteratively optimizing scorer functions of all experts, i.e., $\{g_i\}_{i=1}^C$, we can retrieve the proper $\mathcal{S}(x_{test})$ by Eq. (2) for LLM predictions. We outline the training process in Algorithm 1, with each phase introduced in the following sections.

**Training Data.** The training data construction process is detailed in Phase 1 of Algorithm 1. At the $t$-th epoch, we first sample a batch of samples $d^{(t)} \subset \mathcal{D}$. For each sample $(x_i^{(t)}, y_i^{(t)}) \in d^{(t)}$, we use the scorer functions $\{g_j^{(t-1)}\}_{j=1}^C$ to select the corresponding $\mathcal{S}_-(x_i^{(t)})$ as follows:

$$\mathcal{S}_-(x_i^{(t)}) = \bigcup_{j=1}^{C} \underset{\widehat{\mathcal{S}}_{j-} \subseteq \mathcal{C}_j}{\operatorname{argmax}} \sum_{e \in \widehat{\mathcal{S}}_{j-}} g_j^{(t-1)}(x_i^{(t)}, e), \text{ where } |\widehat{\mathcal{S}}_{j-}| = \lfloor h(\mathcal{C}_j, x_i^{(t)}) * (L-1) \rfloor. \tag{7}$$

For experts that contribute to the prediction for $x_i^{(t)}$, i.e., $|\widehat{\mathcal{S}}_{j-}| > 0$, we use $g_j^{(t-1)}$ to retrieve $K$ candidate demonstrations $\mathcal{E}_j(x_i^{(t)}) = \{e_j^k\}_{k=1}^K$ with the top-$K$ highest scores from the unselected demonstration set $\mathcal{C}_j \setminus \mathcal{S}_-(x_i^{(t)})$ of each expert $j$. The $K$ candidate demonstrations are obtained as follows:

$$\mathcal{E}_j(x_i^{(t)}) = \underset{\mathcal{E} \subset \mathcal{C}_j \setminus \mathcal{S}_-(x_i^{(t)})}{\operatorname{argmax}} \sum_{e \in \mathcal{E}} g_j^{(t-1)}(x_i^{(t)}, e), \text{ where } |\mathcal{E}| = K. \tag{8}$$

These demonstrations will be used as the candidate demonstration set during the following optimization step.

**Few-shot Scoring.** Once we retrieve the top-$K$ demonstrations $\mathcal{E}_j(x_i^{(t)})$ for a sample $(x_i^{(t)}, y_i^{(t)})$ in the batch $d^{(t)}$, we use the criterion $\mathcal{L}$ to score each demonstration for its helpfulness in ICL and use the scores as supervision for optimization. In this work, we employ the log probability of the output as the metric and query the LLM $\mathcal{M}$ for the feedback in the few-shot pattern, i.e., using multiple demonstrations as additional input. For any candidate demonstration $e_j^k, k = 1, 2, \ldots, K$, we score it as

$$s(e_j^k) = \mathcal{L}(y_i^{(t)}, x_i^{(t)}, \{e_j^k\} \cup \mathcal{S}_-(x_i^{(t)})) = \mathcal{P}_{\mathcal{M}}(y_i^{(t)} \mid \{e_j^k\} \cup \mathcal{S}_-(x_i^{(t)}), x_i^{(t)}), \tag{9}$$

which represents the probability of the LLM $\mathcal{M}$ generating the correct prediction sequence, conditioned on the selected demonstrations and the input query. Previous works show that this score serves as a suitable proxy for the utility of a demonstration at inference time [34, 57].

After scoring the $K$ candidate demonstrations, we include the tuple $(x_i^{(t)}, \{e_j^k\}_{k=1}^K, \{s(e_j^k)\}_{k=1}^K)$ in the expert $j$'s training set $\mathcal{D}_j^{train}$ for updating its scoring function at the $t$-th epoch, i.e., $g_j^{(t)}(\cdot)$. We iteratively apply the above process for all samples $(x_i^{(t)}, y_i^{(t)})$ in the sampled batch $d^{(t)}$ and employ contrastive learning for model updates.

**Training Loss.** Our training procedure draws inspiration from the concept of contrastive learning [20] that has proven to be effective when it is necessary to compare the performance of different samples. In our work, each scorer function $g$ comprises two encoders: $\mathcal{M}_d$ for demonstration encoding and $\mathcal{M}_q$ for query input encoding. Both encoders are initialized with the `bert-base-uncased` model [8], and their output vectors represent the embeddings of the sequences. In this section, we detail the training process for expert $j$ as in Phase 2 of Algorithm 1. We omit the subscript $j$ for simplicity.

Given a tuple $(x_i^{(t)}, \{e^k\}_{k=1}^K, \{s(e^k)\}_{k=1}^K)$ for optimizing an expert, we construct its training set by including one positive and $2B - 1$ negative demonstrations, denoted as $(x_i^{(t)}, e_{pos}, e_{neg}^1, e_{neg}^2, \ldots, e_{neg}^{2B-1})$, where $B$ is the batch size. The positive demonstration $e_{pos}$ is

Table 1: The datasets used in experiments and their corresponding tasks. # Train and # Validation denote the numbers of samples during training and validation, respectively. # Demo denotes the average number of demonstrations used in each task during validation. # Expert represents the number of experts used in each task.

| Type | Task | # Train | # Validation | # Demo |
|---|---|---|---|---|
| **Classification** | | | | |
| SST-5 [40] | Sentiment Analysis | 8,534 | 1,101 | 40 |
| MRPC [9] | Paraphrase Detection | 3,668 | 408 | 27 |
| MNLI [51] | Natural Language Inference | 392,568 | 19,647 | 40 |
| QNLI [46] | Natural Language Inference | 104,707 | 5,463 | 27 |
| CMSQA [42] | Commonsense Reasoning | 9,740 | 1,221 | 50 |
| HellaSwag [61] | Commonsense Reasoning | 52,611 | 20,006 | 50 |
| **Generation** | | | | |
| WebQs [3] | Open-Domain QA | 3,778 | 2,032 | 50 |
| GeoQuery [60, 37] | Code Generation | 404 | 280 | 50 |
| NL2Bash [24] | Code Generation | 7,441 | 609 | 43 |
| Break [53] | Semantic Parsing | 44,184 | 7,760 | 28 |
| MTOP [22] | Semantic Parsing | 15,564 | 2,235 | 41 |
| SMCalFlow [2, 58] | Semantic Parsing | 102,491 | 14,751 | 22 |

sampled from top $\widetilde{K}$ demonstrations with largest few-shot scores, denoted as $\mathcal{E}_{pos}$, in the candidate set $\{e^k\}_{k=1}^K$ (thus $\widetilde{K} < K$):

$$\mathcal{E}_{pos} = \underset{\mathcal{E} \subset \{e^k\}_{k=1}^K}{\operatorname{argmax}} \sum_{e \in \mathcal{E}} s(e), \text{ where } |\mathcal{E}| = \widetilde{K}. \tag{10}$$

In this manner, we further filter out the demonstrations with low few-shot scores, indicating that they are not suitable for acting as a demonstration accompanied with other optimal demonstrations in $\mathcal{S}_-$. Negative samples $(e_{neg}^1, e_{neg}^2, ..., e_{neg}^{2B-1})$ include: (i) one hard demonstration $e_{hard} = \operatorname{argmin}_{e \in \{e^k\}_{k=1}^K} s(e)$; (ii) $B - 1$ positive demonstrations from the other $B - 1$ samples in $d^{(t)}$; and (iii) $B - 1$ hard negative demonstrations from those samples. The score returned by $g$ is defined as $g(x, e) = \langle \mathcal{M}_d(e), \mathcal{M}_q(x) \rangle$. We then propose the contrastive learning loss and use it to update $g$:

$$\mathcal{L}(x_i^{(t)}, e_{pos}, e_{neg}^1, e_{neg}^2, ..., e_{neg}^{2B-1}) = -\log \frac{\exp(g(x_i^{(t)}, e_{pos}))}{\exp(g(x_i^{(t)}, e_{pos})) + \sum_{j=1}^{2B-1} \exp(g(x_i^{(t)}, e_{neg}^j))}. \tag{11}$$

Intuitively, the above loss will assign higher scores for demonstrations that are more helpful, when other demonstrations are already optimal. Thus, our expert-wise training could alleviate the impact of unhelpful demonstrations during optimization.

## 3.5 Inference

In the inference stage, we select demonstrations for an input query $x_{test}$ according to Eq. (2), and obtain the prediction $\hat{y} = \operatorname{argmax}_y \mathcal{P}_{\mathcal{M}}(y|\mathcal{S}(x_{test}), x_{test})$ given by LLM $\mathcal{M}$. Although we update the retriever models independently for each expert, each retriever model is designed to select demonstrations that benefit ICL in few-shot scenarios, i.e., using a set of demonstrations as additional input. This is ensured because the supervision scores in Eq. (9) for training the retriever models are generated in a few-shot pattern with a set of demonstrations. For the optimal retriever models $\{g_j^*\}_{j=1}^C$, each model essentially solves the problem: *"Given a good demonstration set $S_-^*$ of size $L - 1$, which demonstration should the expert choose to make the best prediction in $L$-shot ICL?"* Consequently, for any input query, the experts in MoD can collaboratively retrieve a set of demonstrations that could most effectively aid in making accurate predictions.

Table 2: The comparative results of our method and other baselines on various datasets. We present the absolute performance gain over CEIL, and the best results are shown in bold.

| Method | SST-5 | MRPC | QNLI | MNLI | CMSQA | Swag | WebQs | GeoQ | NL2Bash | Break | MTOP | SMCal | Avg. |
|---|---|---|---|---|---|---|---|---|---|---|---|---|---|
| *Learning-free* | | | | | | | | | | | | | |
| Random | 31.43 | 67.65 | 56.67 | 37.74 | 42.51 | 41.16 | 4.87 | 33.93 | 34.35 | 1.70 | 7.30 | 8.90 | 30.68 |
| TopK-BM25 | 36.06 | 69.36 | 62.29 | 40.68 | 36.12 | 42.20 | 16.68 | 62.86 | 58.98 | 26.00 | 52.70 | 46.10 | 45.84 |
| TopK-C | 37.06 | 67.89 | 60.97 | 45.28 | 36.12 | 41.60 | 17.62 | 68.93 | 53.69 | 26.34 | 49.84 | 43.44 | 45.73 |
| TopK-S | 37.06 | 66.91 | 61.58 | 44.85 | 35.54 | 41.69 | 16.83 | 66.43 | 54.89 | 26.58 | 47.29 | 42.59 | 45.19 |
| TopK-BERT | 37.24 | 69.36 | 64.65 | 42.15 | 35.38 | 40.28 | 17.08 | 66.79 | 51.30 | 26.84 | 52.13 | 44.63 | 45.65 |
| *Learning* | | | | | | | | | | | | | |
| EPR | 42.82 | 75.98 | 80.76 | 66.06 | 36.77 | 42.61 | 19.59 | 68.57 | 56.82 | 31.90 | 64.20 | 54.30 | 53.37 |
| CEIL | 47.05 | 80.15 | 85.41 | 71.74 | 37.18 | 43.20 | 20.92 | 73.21 | 59.91 | 34.18 | 67.43 | 60.73 | 56.76 |
| MoD | **48.12** | **81.53** | **86.63** | **73.24** | **43.24** | **44.54** | **21.45** | **73.75** | **62.94** | **35.80** | **69.32** | **62.97** | **58.63** |
| $\Delta$ Gain | +1.07 | +1.38 | +1.22 | +1.50 | +6.06 | +1.34 | +0.53 | +0.54 | +3.03 | +1.62 | +1.89 | +2.24 | +1.87 |

# 4 Experiments

## 4.1 Experimental Settings

**Baselines.** Our MoD framework functions as a mixture of multiple learning-based retrievers for selecting in-context examples from different subsets in the entire training set. We compare it against both learning-free and learning-based retrievers. Learning-free methods include Random, TopK-BM25 [33], TopK-Contriver [17], and TopK-SimCSE [11]. Learning-based methods include EPR [34] and CEIL [57]. We provide more details in Appendix B.2.

**Datasets.** To ensure a fair comparison between our framework and other baselines, following CEIL [57], we conduct experiments on a variety of datasets, involving both classification and generation tasks. For the evaluation on classification datasets, we measure the accuracy of the output regarding the correct answers. For evaluation on generation tasks, we adopt the metrics of Exact Match (EM) scores for all generation datasets except Break, for which we use LF-EM [12] that additionally considers semantic equivalence. Following CEIL [57], we present the final results based on the validation set as test sets are unavailable for specific datasets.

**Implementation Details.** To keep consistency with CEIL [57] and EPR [34], we primarily use GPT-Neo [4], a 2.7-billion-parameter language model trained on The Pile [10], which is an 825GB text corpus collected from various high-quality resources. In Sec. 4.5, we additionally consider three models: GPT2-XL [31] with 1.5 billion parameters, LLaMA-7B [44] with 7 billion parameters, and GPT3.5 [5] with a significantly larger parameter size. The number of in-context demonstrations in our experiments is set as 50, while we truncate this number when the combined length exceeds the maximum context size of LLMs for each task. The ultimate average number of in-context demonstrations used in each task is provided in Table 1. We provide details of the settings in Appendix B.3.

## 4.2 Comparative Results

In Table 2, we report the results of our framework MoD and other baselines on two sets of datasets: six classification datasets and six generation datasets, covering seven tasks. From the results, we could obtain the following observations: (1) **Superior Performance.** MoD demonstrates superior performance across a diverse set of tasks, both in classification and generation, as evidenced by the highest average score (58.63%) compared to competitive baselines CEIL (56.76%) and EPR (53.37%). This indicates that MoD is more effective in leveraging in-context demonstrations to enhance task performance. (2) **Better on Classification.** Compared with CEIL, MoD generally achieves higher performance gain on classification tasks than on generation tasks (Average $\Delta$ Gain 2.10 on classification tasks v.s. Average $\Delta$ Gain 1.64 on generation tasks). This is because our design of the mixture-of-expert architecture enables the selection of demonstrations with a large distance in the embedding space to the query. As classification tasks could be more easily affected by several demonstrations, these selected demonstrations could potentially carry helpful information for inference on the query, while not necessarily being similar to the query in the embedding space. (3)

Table 3: Performance of our framework and various baselines on processed compositional datasets GeoQuery and SMCalFlow-CS. S refers to a non-compositional test set and C refers to a compositional set with additional cross-domain examples as demonstrations.

| Model | GeoQuery | | | | SMCalFlow-CS | |
| | Standard | Template | TMCD | Length | S | C |
|---|---|---|---|---|---|---|
| TopK-BERT | 66.79 | 30.75 | 41.82 | 31.59 | 31.94 | 0.28 |
| EPR | 68.57 | 38.95 | 44.09 | 32.27 | 57.78 | 0.00 |
| CEIL | 73.21 | 40.77 | 44.09 | 32.73 | 60.27 | 0.28 |
| MoD | **77.38** | **41.84** | **44.55** | **33.19** | **62.95** | **0.39** |
| Δ Performance | +4.17 | +1.07 | +0.46 | +0.46 | +2.68 | +0.11 |

**Require Less Data.** MoD's consistent performance from large-scale datasets like MNLI (392,568 training samples) to smaller datasets like GeoQuery (404 training samples) suggests that it effectively generalizes across datasets with varying sizes. The superior performance of MoD on smaller datasets like GeoQuery and NL2Bash demonstrates its ability to learn effectively even with limited labeled data for demonstration selection.

### 4.3 Results on Compositional Datasets

A critical advantage of MoD is its capability to collaboratively select demonstrations from multiple experts, such that these demonstrations are maximally helpful when the other demonstrations in the selected set are also optimal. To evaluate whether the demonstrations retrieved from various experts could be entirely helpful for ICL, we conduct experiments on two semantic parsing datasets derived from the original SMCalFlow and GeoQuery datasets and processed by CEIL [57]. Specifically, the inference on queries in these datasets requires the precise retrieval of multiple specific demonstrations. In other words, without precise retrieval, it is particularly difficult to answer these queries. We provide more details of the dataset settings in Appendix B.1. Following CEIL, we utilize the same trained retriever models of experts as used in Sec. 4.2. From the results presented in Table 3, we could obtain the following observations: (1) The performance of MoD is consistently superior compared to other baselines across datasets. Notably, these tasks require the retrieval of compositional demonstrations that are all important but may not necessarily be similar to each other. In this regard, our proposed MoD framework directly retrieves a diverse set of demonstrations, which significantly enhances the efficacy of few-shot ICL, compared to other basins in this scenario. (2) MoD demonstrates notable improvements on the cross-domain splits (C) of the SMCalFlow-CS dataset. Specifically, MoD achieves gains of +0.11% over CEIL on the cross-domain split. This performance indicates MoD's ability to handle complex, multi-domain tasks by effectively selecting and utilizing diverse in-context examples from multiple experts.

### 4.4 Reduction of ICL Demonstrations

In this subsection, we aim to explore the capability of our MoD framework in scenarios where the number of ICL demonstrations selected from the training set is decreased. This is critical for evaluating the practicality of MoD, as it could be challenging to leverage sufficient demonstrations, due to the lack of data or limitation of model sizes. Particularly, we conduct experiments with different numbers of in-context demonstrations on two classification datasets SST-5 and CMSQA, and two generation datasets GeoQuery and MTOP. We present the performance of MoD over the state-of-the-art baseline CEIL in Fig. 2. From the results, we could observe that particularly on classification datasets SST-5 and CMSQA, our performance improvements over CEIL are more significant. This indicates that for classification tasks that require diverse

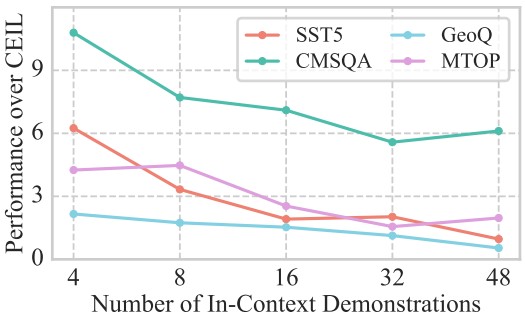

Figure 2: The results of MoD performance over CEIL on various datasets with different numbers of demonstrations. We report the absolute gain of the results.

ments over CEIL are more significant. This indicates that for classification tasks that require diverse

knowledge, our strategy using multiple experts could effectively retrieve crucial demonstrations, which could provide sufficient knowledge even with a limited context length. The performance improvements are relatively consistent on the two generation datasets, i.e., GeoQuery and MTOP. This is because the generation tasks are generally more difficult, and thus require a larger demonstration set. As a result, the advantage of MoD in retrieving diverse knowledge becomes less substantial for model performance.

## 4.5 Robustness Study

In this subsection, we aim to evaluate the robustness, especially the generalizability and transferability of our method MoD to various LLMs. Particularly, our experiments are designed to test whether the retriever models in our MoD framework trained on one LLM could be transferred to other LLMs. Conducting experiments to answer this question could help investigate the applicability of MoD when deployed in realistic scenarios, where LLMs could have different architectures and parameter sizes. Specifically, we use the retriever models trained on GPT-Neo to select demonstrations for the other two models: GPT2-XL with a slightly smaller parameter size and GPT3.5 with a significantly larger parameter size. We present the results of MoD over TopK-BERT in Table 4. From the results, we could observe that (1) The retriever models

Table 4: Performance improvements over TopK-BERT when transferring learned retriever models in MoD to other LLMs on four datasets.

| Model | SST-5 | CMSQA | GeoQ | MTOP |
|---|---|---|---|---|
| **Trained on GPT-Neo** | | | | |
| GPT-Neo | 10.88 | 7.86 | 6.96 | 17.19 |
| GPT2-XL | 8.39 | 8.57 | 6.10 | 15.34 |
| LLaMA-7B | 4.28 | 5.63 | 6.27 | 9.80 |
| GPT3.5 | 3.24 | 6.58 | 4.97 | 7.98 |
| **Trained on LLaMA-7B** | | | | |
| GPT-Neo | 9.67 | 6.92 | 7.34 | 16.05 |
| GPT2-XL | 7.48 | 7.83 | 6.45 | 14.89 |
| LLaMA-7B | 4.12 | 5.47 | 5.10 | 10.27 |
| GPT3.5 | 2.98 | 6.22 | 5.02 | 8.45 |

trained on GPT-Neo exhibit competitive performance when transferred to other LLMs across various datasets. This indicates the transferability of MoD, especially its scalability to large black-box models like GPT3.5. (2) The performance improvements on GPT3.5 are less competitive. This is because due to the powerfulness of GPT3.5, simple methods like TopK-BERT already perform well. Nevertheless, MoD could still improve performance by retrieving better demonstrations. (3) When transferring the retriever models trained on LLaMA-7B to smaller models, the performance improvements are less obvious, probably due to the discrepancy between LLMs in understanding demonstrations.

## 4.6 Ablation Study

In this subsection, we aim to evaluate the specific benefits to performance brought by different modules and designs in our MoD framework. In particular, we evaluate the performance of our MoD framework on four datasets: SST-5, CMSQA, GeoQuery, and MTOP, distinctly covering two classification tasks and two generation tasks. As presented in Fig. 3, we investigate the impact of two key components of our framework: the mixture-of-experts design (MoD w/o E) and the expert-wise training (MoD w/o C). The first variant of our ablation study involves removing the mixture-of-experts design, which results in a significant drop in performance across all datasets, highlighting the importance of leveraging multiple experts for robust prediction.

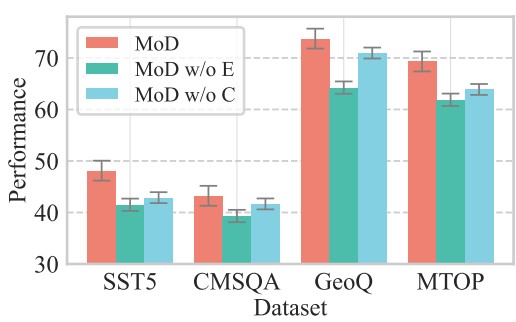

Figure 3: The ablation study result.

The second variant excludes the expert-wise training process, which leads to a moderate decrease in performance, indicating its role in improving the model's performance. Moreover, the results demonstrate that removing the mixture-of-experts design is particularly detrimental for classification tasks, such as SST-5 and CMSQA. Therefore, this underscores its critical contribution to retrieving more diverse and complex demonstrations, which are more crucial for classification tasks.

# 5  Conclusion

In this work, we propose to divide the demonstration retrieval process for in-context learning into multiple parts, each governed by an expert to select from its own sample pool. Our proposed MoD framework further performs expert-wise training to filter out unhelpful demonstrations when optimizing each candidate demonstration. We conduct extensive experiments across a variety of datasets and tasks, and the results validate the superiority of MoD over other baselines.

## Acknowledgments and Disclosure of Funding

This work is supported in part by the National Science Foundation under grants (IIS-2006844, IIS-2144209, IIS-2223769, CNS-2154962, BCS-2228534, CMMI-2411248, CNS-2002902, ECCS-2029978, ECCS-2143559, and CNS-2313110), the Commonwealth Cyber Initiative Awards under grants (VV-1Q24-011, VV-1Q25-004), and the research gift funding from Netflix and Snap.

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

# A Algorithm

We provide the detailed process of expert-wise training in our MoD framework as follows.

---
**Algorithm 1** Expert-wise training

---
1: **Input:** The demonstration pool $\mathcal{D} = \{e_i\}_{i=1}^n = \{(x_i, y_i)\}_{i=1}^n$, Experts' data $\{\mathcal{C}_i\}_{i=1}^C$, large language model $\mathcal{M}$;
2: **Output:** Experts' retriever models $\{g_i\}_{i=1}^C$;
3: Initialize $\{g_j^{(0)}\}_{j=1}^C$ with BERT-base model, initialize experts' training data $\{\mathcal{D}_j^{train}\}_{j=1}^C \leftarrow \{\emptyset\}_{j=1}^C$;
4: **for** $t = 1$ **to** $T$ **do**
5:    ================= Phase 1: Sampling ====================
6:    Batch data sampling $d^{(t)} \in \mathcal{D}$;
7:    **for** $(x_i^{(t)}, y_i^{(t)}) \in d^{(t)}$ **do**
8:       Compute the expert scores $\{h(\mathcal{C}_j, x_i^{(t)})\}_{j=1}^C$;
9:       Retrieve the demonstration subset $\mathcal{S}_-(x_i^{(t)}) = \bigcup_{j=1}^C \widehat{\mathcal{S}}_{j-}^t$ with retriever models $\{g_j^{t-1}\}_{j=1}^C$;
10:       **for** expert $j \in \{1, 2, \ldots, C\}$ and $|\widehat{\mathcal{S}}_{j-}^t| > 0$ **do**
11:          Retrieve the candidate set $\{e_j^1, e_j^2, ..., e_j^K\}$ with top-K score $g_j^{t-1}(x_i^{(t)}, e)$;
12:          Query LLM $\mathcal{M}$ and get the feedback $s(e_j^k) = \mathcal{L}(y_i^{(t)}, x_i^{(t)}, \{e_j^k\} \cup \mathcal{S}_-(x_i^{(t)})), k \in [K]$;
13:          $\mathcal{D}_j^{train} \leftarrow \mathcal{D}_j^{train} \cup \{(x_i^{(t)}, \{e_j^k\}_{k=1}^K, \{s(e_j^k)\}_{k=1}^K)\}$;
14:       **end for**
15:    **end for**
16:    ================= Phase 2: Updating ====================
17:    **for** $j = 1$ **to** $C$ **do**
18:       Update the retriever model $g_j^t$ with training data $\mathcal{D}_j^{train}$ according to Eq. (11);
19:       Empty experts' training data $\mathcal{D}_j^{train} \leftarrow \emptyset$.
20:    **end for**
21: **end for**

---

# B Experimental Settings

## B.1 Datasets

In this work, we evaluate our framework and other baselines on 12 classification and generation tasks. Details for each dataset are summarized below and examples are presented in Table 5.

- **SST-5 [40]:** A fine-grained sentiment classification benchmark with five classes: 'very positive', 'positive', 'neutral', 'negative', and 'very negative'.
- **MRPC [9]:** Determine if two sentences are paraphrases from one another or not.
- **MNLI [51]:** A collection of sentence pairs with textual entailment annotations, where the task is to determine if a sentence entails, contradicts, or is unrelated to a given hypothesis.
- **QNLI [46]:** A NLP inference dataset consists of question-paragraph pairs. The dataset was converted into sentence pair classification by pairing each question with each sentence in the context, then filtering out pairs with low lexical overlap. The task is to determine if the context sentence contains the answer to the question.
- **CMSQA [42]:** Also referred to as CommonsenseQA, this dataset involves multiple-choice questions and necessitates various types of commonsense knowledge to determine the correct answer.
- **HellaSwag [61]:** HellaSwag is a dataset for studying grounded commonsense inference. Each question comes with four answer choices predicting what might happen next in a given scene. The correct answer is the actual subsequent event, while the three incorrect answers are adversarially generated and verified by humans.

- **WebQs [3]:** Also known as WebQuestions, this dataset comprises question-answer pairs sourced from the web. Questions are selected using the Google Suggest API, and the benchmark uses Freebase as the knowledge base.

- **NL2Bash [24]:** The goal of this benchmark is to map sentences to formal Bash commands of their underlying meaning.

- **GeoQuery [60, 37]:** It contains natural language questions about US geography. Shaw et al. [37] further generate multiple splits focusing on compositional generalization. In addition to the original Standard split, it contains three additional splits: (1) the **Template** split, where abstract output templates in training and test data are disjoint; (2) the **TMCD** split, which makes the distributions of compounds in training and test data as divergent as possible; and (3) the **Length** split, where the test instances are longer than the training ones.

- **Break [53]:** Break is a question understanding dataset for complex questions reasoning. It annotates NLP questions with their question decomposition meaning representations. We use the low-level BREAK subset as in previous works [34, 57].

- **MTOP [22]:** A multilingual task-oriented semantic parsing dataset covering 6 languages and 11 domains, which contain compositional representations that allow complex nested queries. We use the English subset of MTOP as in previous works [34, 57].

- **SMCalFlow [2, 58]:** features complex dialogues about events, weather, places, and people. Each dialogue state is represented as a dataflow graph. Its dialog states also feature explicit functions for references and revisions. The SMCalFlow-CS [58] subset consists of single-turn natural language sentences pertaining to two domains: organization structure and event creation, each with its own set of program symbols. The cross-domain (C) test set evaluates examples that incorporate compositional abilities, while the single-domain (S) test set contains examples from a single domain. Due to input length restrictions, we conduct 8-C experiments following CEIL [57], where an additional 8 cross-domain examples are included in the training set to provide composition symbols for evaluation.

## B.2 Baselines

In this subsection, we introduce the details of the baselines used in our framework.

- **RANDOM:** This retriever randomly picks in-context examples from the training set without any repetition.

- **TopK-BM25:** This method employs the classical sparse retrieval technique BM25 [33], an extension of TF-IDF. It selects the top-$K$ scored examples as in-context examples.

- **TopK-BERT:** A dense retriever based on BERT embeddings [8]. Following prevsioue works [57], we use the `bert-base-uncased` model available in Huggingface Transformers [52].

- **TopK-Contriver [17] and TopK-SimCSE [11]:** These are advanced sentence embedding models trained with contrastive learning.

- **EPR [34]:** A learning-based dense retriever trained to find the best singleton in-context example. During the inference stage, it selects the top-K most similar examples.

- **CEIL [57]:** The state-of-the-art baseline instantiated by Determinantal Point Processes (DPPs) to model interactions between the input and demonstrations for in-context learning. It is optimized through a contrastive learning objective with supervision from LMs.

## B.3 Implementation Details

Regarding the experiments in this work, we use a batch size of 128 and a learning rate of $10^{-5}$. We set the size of the candidate demonstration set as $K = 50$. The size of the positive demonstration set is $\widetilde{K} = 10$. We conduct experiments on two NVIDIA A100 GPUs, each with 80GB of memory. For models that are available, we use the implementations provided in Huggingface Transformers [52]. We provide the code at https://github.com/SongW-SW/MoD.

Table 5: Datasets with corresponding prompts and examples used in the experiments.

| Dataset | Prompt | Example |
|---|---|---|
| SST-5 | {input} It is {output} | Input: The film equivalent of a toy chest whose contents get scattered over the course of 80 minutes. Output: Negative. |
| MRPC | {input1} Can we say "{input2}"? {output} | Input1: Gov. Bob Riley proposed the budget cuts after Alabama voters rejected his $ 1.2 billion tax plan Sept . 9. Input2: After Alabama voters rejected his $ 1.2 billion tax plan Sept . 9, Riley forecast significant cuts in state programs. Output: Yes |
| MNLI | {input1} Can we say "{input2}"? {output} | Input1: At 8:34, the Boston Center controller received a third transmission from American 11. Input2: The Boston Center controller got a third transmission from American 11. Output: Yes |
| QNLI | {input1} Can we know "{input2}"? {output} | Input1: Dell continues to remain secretive about their motherboard pin-outs for peripherals (such as MMC readers and power on/off switches and LEDs). Input2: What part of their motherboards does Dell not reveal the specifications of? Output: Yes |
| CMSQA | {input} {output} | Input: If someone laughs after surprising them they have a good sense of what? Output: humor |
| HellaSwag | {input} {output} | Input: The topic is Cleaning sink. A middle-aged female talks about a cleaning product. The female opens a container of cleaner and puts it on a rag. the female, Options: "then inflames a different cleaner to clean a sock.", "uses the rag to spray down a wall.", "washes the rug thoroughly and scratches it.", "then uses the rag to rub the inside of the sink." Output: then uses the rag to rub the inside of the sink |
| WebQs | {input} {output} | Input: what time zone am i in Cleveland, Ohio? Output: North American Eastern Time Zone |
| GeoQuery | {input}\t{output} | Input: What is the area of California? Output: `SELECT state.area FROM state WHERE state.name ='california'` |
| NL2Bash | {input}\t{output} | Input: display the 5 largest files in the current directory and its sub-directories. Output: `find .  -type f | sort -nk 5,5 | tail -5` |
| Break | {input}\t{output} | Input: What is the code of the city with the most students? Output: 1) cities 2) students in #1 3) number of #2 for each #1 4) #1 where #3 is highest 5) code of #4 |
| MTOP | {input}\t{output} | Input: call Zoey's wife. Output: [IN:CREATE_CALL = [SL:CONTACT = [IN:GET_CONTACT = [SL:CONTACT_RELATED = Zoey] [SL:TYPE_RELATION = wife]]]] |
| SMCalFlow | {input}\t{output} | Input: Can you remind me to go to the airport tomorrow morning at 8am? Output: createCommitEventWrapper( createPreflightEventWrapper( EventBuilder( subject='go to the airport', start=dateAtTime( date=tomorrow(), time=numberAM(8))))) |

## C    Technical Details

### C.1    Batch Sampling

At each epoch, our objective is to update all expert's models; therefore, we adopt a stratified sampling strategy to ensure $\mathcal{D}_j^{train}$ is not empty for any expert $j$. Specifically, given the sample fraction $r$, we randomly sample $\max(1, \lfloor r * |\mathcal{C}_j| \rfloor)$ demonstrations from each expert $j$'s demonstration set $\mathcal{C}_j$ and aggregate them to form $d^{(t)}$. This guarantees that each $\mathcal{C}_j$ contributes at least one sample $(x_j, y_j) \in \mathcal{C}_j \cap d^{(t)}$, resulting in $|\widehat{\mathcal{S}}_{j-}^t| > 0$. Consequently, we add $(x_j, \{e_j^k\}_{k=1}^K, \{s(e_j^k)\}_{k=1}^K)$ to $\mathcal{D}_j^{train}$ and make it nonempty.

## D    Complexity Analysis of MoD

We primarily compare the proposed MoD with the state-of-the-art CEIL method [57], focusing on two aspects of complexity reduction: the number of demonstrations used and the efficiency of the inference stage.

**Efficiency of the Number of Demonstrations**    Since the attention mechanism in most LLMs has quadratic complexity [57], fewer demonstrations result in shorter input lengths and reduced computational cost. From Table 6, we observe that MoD generally outperforms CEIL using only 4 demonstrations compared to CEIL's 16 demonstrations. This shows that MoD can achieve better performance with fewer examples, thus reducing the computation complexity in the attention module of LLMs.

Table 6: Performance under various numbers of in-context examples.

| Method | $L$ | MRPC | SST-5 | MTOP |
|--------|-----|-------|-------|-------|
| CEIL   | 4   | 79.28 | 41.25 | 63.40 |
| MoD    | 4   | 80.34 | 47.50 | 67.65 |
| CEIL   | 16  | 79.57 | 46.28 | 65.75 |
| MoD    | 16  | 80.72 | 48.20 | 68.29 |

**Efficiency of the Inference Stage**    As for the inference stage, both MoD and CEIL need to compute the similarity between the query and all $N$ demonstrations, denoted by the complexity as $\mathcal{O}(T)$. CEIL uses a KNN retriever to select $n$ candidates $(n \ll N)$ to narrow the search space. The complexity of selecting top-$n$ candidates is $\mathcal{O}(N + n \log n)$, where $\mathcal{O}(N)$ is to build a max-heap and $\mathcal{O}(n \log n)$ to extract the top-$n$ elements. Then, CEIL uses a greedy algorithm with Cholesky decomposition to reduce the selection complexity from $\mathcal{O}(nL^4)$ to $\mathcal{O}(nL^2)$, where $L$ is the number of ICL examples. Thus, the total complexity of CEIL at the inference stage is $\mathcal{O}(T + N + n \log n + nL^2)$.

In MoD, in the worst case, we select the top $L$ elements in one expert, with a complexity of $\mathcal{O}(N + L \log L)$. Thus, the total complexity of MoD at the inference stage is $\mathcal{O}(T + N + L \log L)$. Given $L < N$, MoD further reduces complexity compared to CEIL at the inference stage.

## E    Additional Experiments

### E.1    Impact of Designs in Expert-wise Training

We conduct experiments focusing on the effect of specific designs in expert-wise training, and the results are reported in Table 7. We consider the following variants: (i) The variant MoD w/o F removes the few-shot scoring strategy, such that the supervision score of each sample is obtained by individually using itself as context. (ii) The variant MoD w/o T alters the strategy of selecting the demonstration set $\mathcal{S}_-(x)$ to random selection, instead of selecting the $L - 1$ highest-scored demonstrations. (iii) The variant MoD w/o N removes the negative demonstrations from other samples in the contrastive learning loss. As a result, the contrastive learning loss only involves

one hard negative sample. We could observe that removing the few-shot scoring strategy causes a significant performance drop. This indicates that it is more suitable to use multiple demonstrations together as input to correctly evaluate the benefit of any demonstration. The results of the other two variants also indicate the importance of using the highest-scored samples as demonstrations and using more negative samples for contrastive loss.

Table 7: Ablation study results of specific designs in the expert-wise training.

| Variant \ Dataset | SST-5 | CMSQA | GeoQ | MTOP |
|---|---|---|---|---|
| MoD w/o F | 44.07 | 41.65 | 71.35 | 64.36 |
| MoD w/o T | 46.42 | 42.69 | 72.77 | 66.89 |
| MoD w/o N | 45.11 | 43.59 | 72.07 | 67.23 |
| MoD | 48.12 | 43.24 | 73.75 | 69.32 |

## E.2 Transferability of MoD Retriever

Regarding the transferability of the retriever in MoD across different tasks, we conduct additional experiments to evaluate the performance of our retriever trained on one dataset and then applied to other datasets. We report the absolute improvement over the baseline TopK-BERT. From the results, we observe a strong pattern that, the performance experiences a reduction when the retriever is transferred to other datasets, indicating that the knowledge in the training dataset is crucial for selecting demonstrations. Moreover, when transferring the retriever from dataset MNLI to other datasets, the performance is decreased greatly. This is potentially due to that the NLI task requires two textual inputs instead of one in other datasets. As such, the learned knowledge in the retriever can hardly be transferred. On the other hand, the performance of our work after transferring is still generally better than TopK-BERT. This verifies the transferability of our work. Developing a retriever that works effectively across all tasks is a challenging yet valuable research topic, which we leave for future work.

Table 8: Results of transferring a retriever learned on one dataset (row) to others (column). We report the absolute improvement over the baseline TopK-BERT.

| Source \ Target | SST-5 | MNLI | GeoQ | MTOP |
|---|---|---|---|---|
| **SST-5** | 10.88 | 7.42 | -1.26 | 0.58 |
| **MNLI** | -4.79 | 31.09 | -13.58 | -31.91 |
| **GeoQ** | 1.42 | 5.98 | 6.96 | 3.46 |
| **MTOP** | 1.37 | 9.08 | 3.80 | 12.56 |

## E.3 Effect of Embedding Models

In this subsection, we investigate the impact of Sentence-BERT on clustering performance, using two variants of Arctic-Embed [28]: Arctic-xs and Arctic-m. We evaluate clustering quality using three metrics: Silhouette Score, Davies-Bouldin Index, and Dunn Index.

As shown in Table 9, Sentence-BERT generally achieves superior clustering results. Notably, previous ICL studies have also utilized Sentence-BERT as an embedding model [34, 7, 57]. Our results demonstrate that MoD consistently outperforms other baselines when using the same embedding model. Additionally, we observe that the Dunn Index is more closely correlated with the final performance of ICL. Selecting the appropriate clustering criteria and optimal embedding model for ICL is a challenging yet valuable problem, which we leave for future work.

Table 9: Impact of different embedding models on clustering performance on dataset MRPC.

| Metric | Sentence-BERT | Arctic-xs | Arctic-m |
|---|---|---|---|
| Silhouette Score ↑ | 0.15 | 0.11 | 0.01 |
| Davies-Bouldin Index ↓ | 2.07 | 2.31 | 6.49 |
| Dunn Index ↑ | 0.12 | 0.04 | 0.19 |
| Accuracy | 81.53 | 77.26 | 81.87 |

## E.4 Effect of Retriever Models

We conduct experiments to investigate the influence of different retriever model structures. Note that EPR [34] can be seen as the implementation of DPR [20] for ICL tasks. In Table 10, we present the results of MoD and EPR under different retriever models. The results indicate that replacing the BERT-base model with RoBERTa [33] or DeBERTa [14] enhances the performance of both EPR and MoD in most cases, with MoD consistently outperforming EPR across all retriever models. This suggests that retriever performance can indeed benefit from the choice of encoder model.

Table 10: Impact of different retriever backbone models.

| Method | SST-5 | CMSQA | GeoQ | MTOP |
|---|---|---|---|---|
| EPR | 42.82 | 36.77 | 68.57 | 64.20 |
| EPR w/ RoBERTa | 43.65 | 36.62 | 69.52 | 66.80 |
| EPR w/ DeBERTa | 44.21 | 37.85 | 69.38 | 64.57 |
| MoD | 48.12 | 43.24 | 73.75 | 69.32 |
| MoD w/ RoBERTa | 49.41 | 44.12 | 74.52 | 70.61 |
| MoD w/ DeBERTa | 49.13 | 43.20 | 74.90 | 71.46 |

## E.5 Effect of $K$ and $\tilde{K}$

We present the results for different values of $K$ and $\tilde{K}$ in Table 11. The results indicate that increasing the value of $K$ can slightly enhance performance but at the cost of significantly higher computational overhead. Notably, for larger values of $K$, such as $K = 100$, increasing $\tilde{K}$ may inadvertently degrade performance. This decline is likely due to the inclusion of positive demonstrations with relatively lower scores as $\tilde{K}$ increases.

Table 11: Effect of $K$ and $\tilde{K}$.

| $K$ \ $\tilde{K}$ | 20 | 10 | 5 |
|---|---|---|---|
| 100 | 45.75 | 48.40 | 48.02 |
| 50 | 46.34 | 48.12 | 47.94 |
| 20 | 47.21 | 47.04 | 47.32 |
| 10 | - | 46.39 | 46.88 |

## E.6 Effect of Hard Negative Sampling

We investigate the effect of hard negative sampling. In the original setting, we set #Hard = 1. In Table 12, we present the results for four variants: #Hard = 1, 5, 10, and 20. Across all datasets, we observe a general trend where performance initially improves with a slight increase in the number of hard negatives, but then begins to decline as the number continues to increase. This pattern suggests

that using a moderate number of hard negative samples strikes a balance between leveraging enough information from negative samples and avoiding the inclusion of potentially irrelevant data.

Table 12: Effect of the number of hard negatives.

| Variant | SST-5 | CMSQA | GeoQ | MTOP |
|---|---|---|---|---|
| MoD #Hard=1 | 48.12 | 43.24 | 73.75 | 69.32 |
| MoD #Hard=5 | 48.45 | 43.79 | 74.12 | 69.53 |
| MoD #Hard=10 | 47.98 | 43.27 | 73.91 | 68.93 |
| MoD #Hard=50 | 47.02 | 42.42 | 72.37 | 67.57 |

## F Limitation Discussion

Our framework MoD aims to select suitable demonstrations to improve the ICL performance of LLMs. However, there still exist limitations to our framework. First, our MoD framework requires the label of samples to provide supervision information to the LLMs. This drawback is also present in recent works such as EPR [34] and CEIL [57]. In the future, it is potentially inspiring to develop a framework that does not require the labels of the demonstrations, i.e., using unlabeled samples. Second, the performance of our MoD framework is related to the assignment of experts. If an input query has misinformation and is assigned to incorrect experts, the retrieved samples from these experts may not be helpful and contribute to the performance.

## G Broader Impacts

In this paper, we propose a demonstration selection approach MoD which aims to select the proper demonstrations as in-context learning prompts to improve the performance of the large language model. The proposed method sheds light on the future design of new and fancy demonstration selection methods. While we emphasize the importance of responsible use, we do not anticipate any major negative societal impacts from our work.

