# OpenReview forum: "Mixture of Demonstrations for In-Context Learning"
_NeurIPS.cc/2024/Conference — NeurIPS 2024 poster_

### Official Review · Reviewer_UC6f · 2024-07-07

**Soundness:** 3
**Presentation:** 2
**Contribution:** 2
**Rating:** 6
**Confidence:** 5

**Summary:**

This paper proposes MoD (Mixture of Demonstrations), which partitions the demonstration pool into clusters using embeddings to enhance the diversity of demonstrations. For each cluster, an individual retriever is trained with the few-shot scoring from the LLM to maximize the combinational effect of the selected demonstrations. During inference, each retriever will retrieve a certain number of the demonstrations based on the similarity with the input query. MoD can outperform previous demonstration selection methods, and the selected demonstrations can generalize to larger models.

**Strengths:**

- It is reasonable to train multiple experts (retrievers) to collaboratively select demonstrations while eliminating the need to enumerate all the combinations in the search place
- The training data (positives and negatives) of the retriever can reflect the model's needs for the demonstrations
- The proposed method achieves stronger performance compared with the baselines adopted in this work
- In general, the paper is easy to read

**Weaknesses:**

- The efficiency of this method may limit its application since it requires training the retriever of each expert iteratively. I would like to see the **total FLOPs** of MoD compared to previous work
- Missing important hyperparameters analysis and ablation studies, e.g.
    - How many iterations are needed to train the retriever well?
    - How does Sentence-BERT affect the cluster performance compared to other embedding models like Arctic-Embed?
    - How does BERT-base affect the retriever performance compared to other retrievers like DPR?
    - How do $K$ and $\widetilde{K}$ affect the retriever's training?
    - How does the hard negative sampling affect the retriever's training? Right now, it is just the $argmin$ score, can we have more hard negatives?
- The function $h$ to determine the relevance between input and expert is just the cosine similarity. Why should we assign more demonstrations to the more similar expert, and any justification?
- In Table 4, the retriever trained by LLaMA-7B feedback cannot **benefit LLaMA-7B more** than the one trained by GPT-Neo feedback, which may pose the question of the usefulness of the model-aware few-shot scoring signal

[1]: Merrick, Luke, et al. "Arctic-Embed: Scalable, Efficient, and Accurate Text Embedding Models." arXiv preprint arXiv:2405.05374 (2024).

[2]: Karpukhin, Vladimir, et al. "Dense Passage Retrieval for Open-Domain Question Answering." Proceedings of the 2020 Conference on Empirical Methods in Natural Language Processing (EMNLP). 2020.

**Questions:**

- Why the gain over CEIL on the cross-domain split of SMCalFlow-CS is only 0.11%? Is it because the task is extremely hard?
- Why does Table 4 show the MoD performance over TopK-BERT (a weaker baseline), not CEIL?
- Can you provide some cases of good vs. bad demonstrations?

---

 Some suggestions:

- $\mathcal{L}$ is repetitively used as the evaluation criterion as well as contrastive loss
- In Table 1, I cannot find *# Expert* that represents the number of experts used in each task
- *basins* typo in line 285
- More concise notations for the equations

**Limitations:**

Yes.

---

> ### Author Rebuttal · Authors · 2024-08-07
>
> > **W1**: Comparison of total FLOPs of MoD compared to previous work.
>
> **RW1**: Hereby we compare the FLOPs of MoD with the SOTA baseline CEIL. We follow the same hyperparameter and encoder settings as CEIL. Thus, **the total FLOPs is linear to the cost of each training sample**. Denote the FLOPs of processing each sample as $\tau$. MoD involves $L-1$ ($L=50$) demonstrations and $C=5$ (on average) experts for each training sample, each with $K=50$ candidate demonstrations. Thus, the computation cost for each training sample is $(CK+L-1)\tau=(5\times 50+50-1)\tau=299\tau$. CEIL selects 50 candidate subsets with 16 examples in each subset for each training sample, which is $50\times16\tau=800\tau$. Hence, our FLOPs is approximately 3/8 of CEIL. We will provide more detailed comparisons in the revised version.
>
> > **W2**: Ablation studies.
>
> **RW2**: Thank you for bringing up this point. We provide the results in Tables 3-6 in the attached PDF. We address each aspect in detail:
> - **Iteration Setting**: we follow the settings as CEIL and set the iteration number $T=30$
> - **Sentence-BERT Effect**: We follow your instructions and conduct experiments with two variants of Arctic-Embed [1]: Arctic-xs and Arctic-m. We evaluate the clustering performance with three metrics: Silhouette Score, Davies-Bouldin Index, and Dunn Index.
>
>      From Table 3 in the attached PDF, we observe that **Sentence-BERT generally achieves superior clustering results**. Previous ICL studies have also utilized Sentence-BERT as an embedding model [2,5,6,8]. Our main results demonstrate that MoD consistently outperforms other baselines with the same embedding model. Furthermore, the Dunn index in Table 3 appears to correlate more closely with the final performance of ICL. Choosing clustering criteria and the optimal embedding model for ICL is a challenging yet valuable task, which we leave to future work.
> - **BERT-base Effect**: Note that the baseline EPR [2] follows the same training process as DPR [3], including the retriever setting and contrastive loss. Thus, EPR can be seen as the implementation of DPR for ICL tasks. In Table 4 in the attached PDF, we present the results of MoD and EPR under different retriever models.
>
>    The results indicate that **replacing the BERT-base model with RoBERTa or DeBERTa enhances the performance of both EPR and MoD in most cases**, with **MoD consistently outperforming EPR across all retriever models**. That means retriever performance can indeed benefit from the choice of encoder models.
>
> - **$K$ and $\tilde{K}$ Effect**: We present the results with different values of $K$ and $\tilde{K}$ in Table 5 in the attached PDF.
>
>    From the result, we note that **improving the value of $K$ could slightly improve the performance**. However, it also brings significantly higher computational costs. Notably, when using a larger value of $K$, e.g. 100, increasing the value of $\tilde{K}$ can advertently decrease the performance.  This is potentially because the $\tilde{K}$ positive demonstrations may involve demonstrations with relatively lower scores when $\tilde{K}$ increases.
> - **Hard Negative Effect**: In Table 6 in the attached PDF, we present results with three variants: \#Hard=1,5,10,20. Note that in the original setting, we set \#Hard=1.
> Across all datasets, **there is a general trend where performance improves initially with a slight increase in the number of hard negatives but begins to decline as it continues to increase**. This pattern suggests that using a moderate number of hard negative samples provides the balance between leveraging enough information from negative samples and avoiding the inclusion of potentially irrelevant data.
>
> > **W3**: Justification of demonstration assignment.
>
> **RW3**: We appreciate the reviewer’s question. We want to justify our approach as follows:
> - **Use of Cosine Similarity**: Han et al. [4] theoretically show that LLMs perform ICL similar to kernel regression. This involves using a kernel $K(f(x),f(y))$ with an embedding function $f$ to measure the similarity between two demonstrations $x$ and $y$. Existing works typically use Sentence-BERT as $f$ and employ cosine similarity to measure semantic similarity [5,6]. Previous research also empirically shows that ICL can benefit from instances that are semantically similar to the query [7].
> - **Assigning Demonstrations to Similar Experts**: After the clustering process, each expert contains semantically similar demonstrations. Given a new query, assigning more demonstrations to the more similar expert ensures that more similar instances are selected. This selection process improves the relevance of the prompts, thereby improving overall performance.
>
> > **W4**: Usefulness of the model-aware signal.
>
> **RW4**:  We appreciate the reviewer for pointing out this observation. We would like to address this concern by highlighting a few key points.
> - **Suitability of LLaMA-7B for Feedback Selection**: It is possible that LLaMA-7B may not be as suitable for feedback selection compared to GPT-Neo. As shown in Table 7 in the attached PDF, the average absolute gain of using GPT-Neo feedback compared to using LLaMA-7B feedback across four tasks is consistently greater than 0. The retriever trained with feedback from a model like GPT-Neo appears to be more adaptable and beneficial across different LLMs.
> - **Model-Aware Signal and Performance Enhancement**: Besides GPT-3.5, which is notably strong, the average absolute transfer gain on LLaMA-7B is the smallest, whereas the average absolute transfer gain on GPT-Neo is the largest. This indicates that the model-aware signal can further enhance ICL performance on the same LLM.
>
> Due to the rebuttal limitation, we include responses to questions and references in the additional comment. Your review would be greatly appreciated.

---

> ### Author Response · Authors · 2024-08-07
> **Author's Responses to Questions**
>
> >**Q1**: Why the gain over CEIL on the cross-domain split of SMCalFlow-CS is only $0.11\%$? Is it because the task is extremely hard?
>
> **RQ1**:  We appreciate the reviewer for mentioning this point. As discussed in Appendix B.1, SMCalFlow is a large-scale semantic parsing dataset for task-oriented dialogue. The cross-domain (C) test set evaluates examples that incorporate compositional skills in dialogue, which is significantly more challenging compared to the single-domain (S) evaluation. Such cross-domain tasks are particularly difficult for LLMs like GPT-Neo. A similar observation is also observed in [8].
>
> However, despite the difficulty of the task, MoD demonstrates a substantial improvement over CEIL. Specifically, **the relative performance gain of (0.39 - 0.28) / 0.28 = 39.3\% indicates that MoD can better handle complex, multi-domain tasks**. This improvement highlights the capability of MoD to effectively address the challenges posed by cross-domain evaluations.
>
> >**Q2**: Why does Table 4 show the MoD performance over TopK-BERT (a weaker baseline), not CEIL?
>
> **RQ2**: Thank you for your question regarding the setting of results in Table 4. We follow the settings in CEIL and thus report the performance over TopK-BERT as an evaluation of the transferability of MoD. Our experimental results in various sections, all demonstrate the superiority of MoD over the baseline CEIL. By including TopK-BERT in our comparisons, we provide a broader context for evaluating the improvements brought by MoD.
>
> >**Q3**: Examples of good vs. bad demonstrations.
>
> **RQ3**:  Thank you for your feedback. Here, we select one training sample along with two good and two bad demonstrations.
> - **Training Sample**: "nicholson's understated performance is wonderful"
> - **Good Demonstrations**:
> 1. "britney's performance cannot be faulted"
> 2. "parris' performance is credible and remarkably mature"
> - **Bad Demonstrations**:
> 1. "there is something that is so meditative and lyrical about babak payami's boldly quirky iranian drama secret ballot..."
> 2. "the film fearlessly gets under the skin of the people involved..."
>
> Notably, the good demonstrations are both semantically and syntactically similar to the training sample. Conversely, the bad demonstrations are not directly opposite in meaning. This indicates that such demonstrations could incur more negative effects than those with straightforward adversarial meanings. We will include more examples in the revised version.
>
> **Suggestions**: We thank the reviewers for their careful reading and subtle observations. We will be happy to make the corresponding changes in subsequent versions for these suggestions.
>
> [1] Merrick et al. Arctic-Embed: Scalable, Efficient, and Accurate Text Embedding Models. Arxiv, 2024.
> [2] Rubin et al. Learning to retrieve prompts for in-context learning. NAACL, 2022.
> [3] Karpukhin et al. "Dense Passage Retrieval for Open-Domain Question Answering." EMNLP, 2020.
> [4] Han et al. In-context learning of large language models explained as kernel regression. Arxiv, 2023.
> [5] Wang et al. One Prompt is not Enough: Automated Construction of a Mixture-of-Expert Prompts. ICML, 2024.
> [6] Su et al. Selective Annotation Makes Language Models Better Few-Shot Learners. ICLR, 2023.
> [7] Liu et al. What makes good in-context examples for gpt3? DeeLIO, 2022.
> [8] Ye et al. Compositional exemplars for in-context learning. ICML, 2023.

---

> > ### Comment · Reviewer_UC6f · 2024-08-13
> > **Thanks for your rebuttal**
> >
> > Thanks to the authors. I have read the rebuttal carefully. I would increase my score from 5 to 6 since the authors addressed most of my concerns reasonably and conducted necessary ablation studies. I hope they can incorporate those better in the next version.

---

> > > ### Author Response · Authors · 2024-08-13
> > > **Response to Reviewer's Reply**
> > >
> > > Dear Reviewer UC6f,
> > >
> > > Thank you for carefully reading our rebuttal. We appreciate your recognition of our efforts to address your concerns.
> > >
> > > We will incorporate your suggestions and further improvements in the revised version. Your input has been invaluable in helping us refine our research.
> > >
> > > Thank you once again for your constructive feedback.
> > >
> > > Best,
> > > The Authors

---

### Official Review · Reviewer_SZXS · 2024-07-11

**Soundness:** 3
**Presentation:** 3
**Contribution:** 2
**Rating:** 6
**Confidence:** 3

**Summary:**

The paper presents a new method for creating a dense retriever for ICL for LLMs performing tasks. The proposed combines a mixture of experts model, coordinate descent (for alternating optimization to the experts), and contrastive loss to create the retriever. The paper then goes on to show the method's performance on creating a retriever across several benchmark datasets and tasks, where it performs better than previous models.

**Strengths:**

The paper has a strong evaluation and is original in its diagnoses of the problems with retrieving for ICL and its proposed solution. The paper has a novel combination of techniques to produce the retriever for ICL. These techniques, like MoE are ideally suited to tackle some of the issues the paper identifies with the current generation of learned retrievers. Additionally, the empirical results bear out the utility of this approach; the MoD retriever produces better in-domain ICL results than any other retriever across a number of LLM tasks, which go beyond just classification tasks. I also appreciate that the paper considered the cross-LLM performance of the trained retriever. Finally, I also appreciate that the paper has many details, like the actual code and the prompting schemes available for the reader.

**Weaknesses:**

For one, the method still requires labeled data in the validation set in order to work. The value of LLMs, particularly for Labeling tasks, is their ability to work in zero-shot settings (for example, see Liyanage et al. “GPT-4 as a Twitter Data Annotator: Unraveling Its PerformanceGPT-4 as a Twitter Data Annotator: Unraveling Its Performance on a Stance Classification Task” and/or Ziems et al. “Can Large Language Models Transform Computational Social Science”). Thus, having to have labeled data examples to get labeled data examples lowers the significance of the proposed method.

Second, the focus of the paper could be better to make the method more significant. The paper focuses on in-domain data and tasks, especially with regard to labeling tasks (e.g., sentiment labeling). While I certainly agree that in-domain and task performance is important, this could really be handled by standard supervised approaches with smaller models. Rather, I think this method should be focused on something like creating new, labeled datasets, based on benchmark datasets. Consider the example in Cruickshank and Ng “DIVERSE: Deciphering Internet Views on the U.S. Military Through Video Comment Stance Analysis, A Novel Benchmark Dataset for Stance Classification”. I think the proposed method could actually work in this type of scenario where you use benchmark stance-labeled datasets like SemEval2016 and srq, etc. to train a retriever, which could then be used with ICL to produce a labeled dataset, without human labeling. In other words, I think the significance of the method, and by extension, the paper, could be substantially improved if they framed the method around more significant problems in using LLMs to label data as opposed to framing it in-domain, supervised performance.

**Questions:**

-	How transferable to different data domains or tasks is a retriever trained in the proposed approach? For example, how well will a retriever trained for ICL on SST-5 work on ICL for MNLI?

**Limitations:**

The authors should mention that the use of LLMs for labeling data, especially for in-domain usage, could be considered a waste of resources. For example, if I am trying to label data for stance and I have a benchmark, labeled dataset that is similar already, I can likely just as good of performance, with much less computational resources training a small, supervised model on the benchmark dataset and applying it to the dataset to be labeled, versus using a computationally expensive LLM to do the same task.

---

> ### Author Rebuttal · Authors · 2024-08-07
>
> >**W1**: The method still requires labeled data in the validation set in order to work. The value of LLMs, particularly for Labeling tasks, is their ability to work in zero-shot settings. Thus, having to have labeled data examples to get labeled data examples lowers the significance of the proposed method.
>
> **RW1**: Thank you for pointing out this. Zero-shot labeling is indeed a very interesting and significant research area. However, our paper focuses on in-context learning, which is also important but represents a different branch compared to zero-shot learning.
>
> In-context learning aims to improve LLM performance given some labeled examples. In contrast, **zero-shot labeling typically requires much larger LLMs**, such as GPT-4, to be effective. Smaller LLMs generally lack this capability, as the labeling task often requires chain-of-thought (CoT) abilities [1], which smaller models do not possess (CoT only yields performance gains when used with models of $\sim$ 100B parameters [2]). **Our method can have a broad impact, particularly for smaller LLMs, by enhancing performance through the selection of high-quality examples**. In other words, when the LLM is relatively small and lacks strong zero-shot labeling capabilities, our method can significantly enhance the model's labeling performance by selecting high-quality samples as prompts. Therefore, we believe that our method has a wide range of impacts.
>
> We acknowledge that the reviewer's point is valid and that zero-shot labeling is a valuable and intriguing research topic. We will focus on this in future work and explore ways to integrate zero-shot capabilities into our approach.
>
> >**W2**: The focus of the paper could be better to make the method more significant. The paper focuses on in-domain data and tasks, especially with regard to labeling tasks (e.g., sentiment labeling). While I certainly agree that in-domain and task performance is important, this could really be handled by standard supervised approaches with smaller models...
>
> **RW2**: Thank you for recognizing the contribution of this work to in-domain ICL, and we appreciate your insightful suggestions. We focus on in-domain data and tasks in order to **ensure consistency with previous works and to demonstrate the efficacy of our method in a controlled setting**.  While supervised approaches with LLMs can also be effective, our ICL method provides a solution without requiring LLM fine-tuning. Our intention is to explore the potential of ICL to enhance labeling performance by selecting high-quality examples. We also want to mention that in Table 3, we investigate the cross-domain generalization ability of the learned retriever. SMCalFlow-CS is a large-scale semantic-parsing dataset for task-oriented dialogue. The cross-domain (C) test set evaluates examples that incorporate compositional skills in dialogue, which is significantly more challenging compared to the single-domain (S) evaluation. MoD demonstrates a substantial improvement over CEIL. Specifically, the performance gain of (0.39 - 0.28) / 0.28 = 39.3\% indicates that **MoD can better handle complex, cross-domain tasks**. This improvement highlights the capability of MoD to effectively address the challenges posed by cross-domain evaluations.
>
> We appreciate the suggestion to apply our method to scenarios such as creating labeled datasets based on benchmark datasets like SemEval2016. This is an exciting direction that could demonstrate the broader impact and utility of our approach. Using benchmark stance-labeled datasets to train a retriever, which could then be used with in-context learning to produce labeled datasets without human intervention, is a compelling application. We will certainly consider incorporating this perspective in our future work.
>
>
> >**Q1**: How transferable to different data domains or tasks is a retriever trained in the proposed approach? For example, how well will a retriever trained for ICL on SST-5 work on ICL for MNLI?
>
> **RQ1**: We appreciate your insightful question regarding the transferability of the retriever in MoD across different tasks. To address this concern, we have conducted additional experiments to evaluate the performance of our retriever trained on one dataset and then applied to other datasets. We report the absolute improvement over the baseline TopK-BERT.
>
> |       | SST5  | MNLI  | GeoQ  | MTOP  |
> |-------|-------|-------|-------|-------|
> | **SST5**  | 10.88  | 7.42  | -1.26 | 0.58 |
> | **MNLI** | -4.79 | 31.09 | -13.58| -31.91|
> | **GeoQ**  | 1.42 | 5.98  | 6.96  | 3.46  |
> | **MTOP**  | 1.37 | 9.08  | 3.80  | 12.56  |
>
> From the results, we observe a strong pattern that, the performance experiences a reduction when the retriever is transferred to other datasets, indicating that **the knowledge in the training dataset is crucial for selecting demonstrations**. Moreover, when transferring the retriever from dataset MNLI to other datasets, the performance is decreased greatly.  This is potentially due to that the NLI task requires two textual inputs instead of one in other datasets. As such, the learned knowledge in the retriever can hardly be transferred. On the other hand, **the performance of our work after transferring is still generally better than TopK-BERT**. This verifies the transferability of our work. Developing a retriever that works effectively across all tasks is a challenging yet valuable research topic, which we leave for future work.
>
> [1] Liyanage et al. GPT-4 as a Twitter Data Annotator: Unraveling Its PerformanceGPT-4 as a Twitter Data Annotator: Unraveling Its Performance on a Stance Classification Task.
>
> [2] Wei et al. Chain-of-thought prompting elicits reasoning in large language models. NeurIPS, 2022

---

> > ### Comment · Reviewer_SZXS · 2024-08-13
> > **Reply to Rebuttal**
> >
> > I want to thank the authors for considering all of the points I brought up. I especially would like to commend them on adding in the cross-dataset test. While I suspected training on one dataset and then retrieving on another would see a performance, it is good validation to see this result. And, even cross-dataset fine-tuning for the retrievers still gives a performance increase over a baseline, as noted by the authors. I believe there is merit in the paper and having validation for difficulties like cross-dataset performance when fine-tuning, and so I stand by my rating for the article.

---

> ### Author Response · Authors · 2024-08-14
> **Official Comment by Authors**
>
> Dear Reviewer SZXS,
>
> Thank you for your thoughtful feedback and for acknowledging our efforts to address your points. We greatly appreciate your recognition of our work's contributions.
>
> We agree that expanding into zero-shot labeling and data generation tasks could significantly enhance the impact of our approach. We will focus on these topics in future work.
>
> Best,
> The Authors

---

### Official Review · Reviewer_SdUm · 2024-07-13

**Soundness:** 2
**Presentation:** 2
**Contribution:** 3
**Rating:** 5
**Confidence:** 3

**Summary:**

The paper introduces the Mixture of Demonstrations (MoD) framework for the demonstration retrieval for In-Context Learning (ICL) in Large Language Models (LLMs). Current methods for selecting demonstrations are hindered by large search spaces and insufficient optimization. MoD addresses these issues by partitioning the demonstration pool into groups managed by experts, optimizing the retriever model to prioritize beneficial demonstrations by coordinate descent, and aggregating expert-selected demonstrations during inference. Experiments across various NLP tasks show MoD's state-of-the-art performance.

**Strengths:**

1. Training strategy: The authors propose a novel training strategy drawing inspiration from coordinate descent
2. Comprehensive Experiments: The paper provides extensive experimental results demonstrating the effectiveness of MoD across a variety of NLP tasks.
3. State-of-the-Art Performance: The MoD framework achieves better performance compared to existing baselines, indicating its practical value.

**Weaknesses:**

1. Lack of complexity analysis. One of main motivation of this paper is to reduce the searching space and complexity while [1] also aims to solve this problem. However, the authors did not quantitatively analyze the complexity of their methods and compared with [1]. Further analysis on the search space complexity and the training cost (espically how many training samples needed to be annotated based on LLM feedback for each task?).
2. Coordinate descent. The paper mentions the use of coordinate descent but does not clearly specify where and how it is applied within the framework. Clarification on the application and role of coordinate descent in the proposed method is needed.
3. The authors claim that MoE enables the retrieval of dissimilar samples that could be beneficial for ICL (lines 123-124). However, [1] and other works achieve this by maximizing the similarity between demonstration examples and minimizing the similarity within the demonstration sequence to introduce diversity. More analysis and comparison of these methods are necessary to verify the effectiveness of MoE.
4. More ablations study. The paper's readability can be improved by explaining each subscript in the equations immediately upon introduction. This would help readers follow the mathematical formulations more easily.

[1] J. Ye, Z. Wu, J. Feng, T. Yu, and L. Kong. Compositional exemplars for in-context learning. In International Conference on Machine Learning, pages 39818–39833. PMLR, 2023.

**Questions:**

Please refer to the weakness.

**Limitations:**

Yes

---

> ### Author Rebuttal · Authors · 2024-08-07
>
> > **W1**: Lack of complexity analysis.
>
> **RW1**: We would like to provide the following analysis of search space and complexity.
>
> - **Search Space**: CEIL precomputes a small relevance set using a pre-trained retriever and optimizes the search process within this narrowed space. While this approach reduces search space, **it can lead to suboptimal demonstration sets due to limited exploration of the entire pool**.
>
>     In contrast, our proposed MoD leverages the MoE mechanism to partition the demonstration pool into subspaces, each managed by an expert. Given an input query, **MoD has the potential to search the entire demonstration pool by retrieving examples from all subspaces**. This ensures more comprehensive exploration.
>
> - **Computation Complexity**: CEIL primarily discusses the reduction of complexity in two ways: **the number of demonstrations** and **the inference stage**.
>
>     Since the attention mechanism in most LLMs has quadratic complexity, fewer demonstrations result in shorter input lengths and reduced computational cost. From Table 1 in the attached PDF, we observe that MoD generally outperforms CEIL using only 4 demonstrations compared to CEIL's 16 demonstrations. This shows that **MoD achieves better performance than CEIL with fewer examples**, resulting in less computation complexity compared to CEIL..
>
>     As for the inference stage, both MoD and CEIL need to compute the similarity between the query and all $N$ demonstrations, denoted the complexity as $\mathcal{O}(T)$. CEIL uses a KNN retriever to select $n$ candidates $(n<<N)$ to narrow the search space. The complexity of selecting top-$n$ candidates is $\mathcal{O}(N+n\log n)$, where $\mathcal{O}(N)$ is to build a max-heap and $\mathcal{O}(n\log n)$ to extract the top-$n$ elements. Then, CEIL uses a greedy algorithm with Cholesky decomposition to reduce the selection complexity from $\mathcal{O}(nL^4)$ to $\mathcal{O}(nL^2)$, where $L$ is the number of ICL examples. Thus, the total complexity of CEIL at the inference stage is $\mathcal{O}(T+N+n\log n+nL^2)$.
>
>     In MoD, in the worst case, we select the top $L$ elements in one expert, with a complexity of $\mathcal{O}(N+L\log L)$. Thus, the total complexity of MoD at the inference stage is $\mathcal{O}(T+N+L\log L)$. Given $L<n$, **MoD further reduces complexity compared to CEIL at the inference stage**.
>
>     Regarding the samples required for each training sample, our framework involves $L-1$ ($L=50$) demonstrations and $C=5$ (on average) experts for each training sample, each with $K=50$ candidate demonstrations. Therefore, the required samples for each training sample is $(CK+L-1)=5\times 50+50-1=299$. In comparison, CEIL selects 50 candidate subsets with 16 examples in each subset for each training sample, which is $50\times16=800$. Hence, our cost is approximately 3/8 of CEIL. We will provide more details in the revised version.
>
> > **W2**: Coordinate descent.
>
> **RW2**: Thank you for highlighting this point, and we would like to clarify our approach. While we do not directly apply coordinate descent, **we draw inspiration from it to manage the large search space and significant optimization overhead**, as mentioned in lines 156-159.
>
> To reduce the computational burden, we iteratively fix most selected demonstrations and focus on optimizing one at a time, which is conceptually related to coordinate descent. Our expert-wise training strategy involves learning the retrieval score for any candidate demonstration by pairing it with demonstrations selected by all experts, which remain fixed while we optimize one candidate at each step. This iterative optimization helps us find the best combination of demonstrations. We will clarify this in the revised version of the paper.
>
> > **W3**: Similarity and diversity.
>
> **RW3**: We appreciate your insights and would like to provide further analysis and comparison of MoD and CEIL.
>
> Our approach, like previous works, **ensures both diversity of demonstrations and their similarity to the input query**. Each expert contains semantically similar demonstrations, represented by their average embedding. Our design of assigning more demonstrations to the most similar expert ensures more relevant demonstrations for the query. For diversity, our MoE mechanism ensures contributions from all experts, not just one.
>
> On the other hand, CEIL also addresses similarity and diversity, but faces the challenge of a massive search space. It precomputes a small set of relevant examples, which can limit diversity and lead to suboptimal selections. In contrast, our method MoD overcomes this by partitioning the space into subspaces, allowing each expert to retrieve relevant demonstrations within its subspace. This enables a thorough search for high-quality demonstrations, leading to improved performance.
>
> > **W4**: Readability and ablation study.
>
> **RW4**: We appreciate your suggestion about the readability. We will improve it in the revised version to enhance clarity. Note that we already have an ablation study in Section 4.8. In Table 2 in the attached PDF, **we provide more ablation results, focusing on the effect of specific designs in the expert-wise training**. We could observe that removing the few-shot scoring strategy causes a significant performance drop. This indicates that it is more suitable to use multiple demonstrations together as input to correctly evaluate the benefit of any demonstration.
> The results of the other two variants also indicate the importance of using the highest-scored samples as demonstrations and using more negative samples for contrastive loss. We also provide more ablation studies of other factors in Tables 3-6 in the attached PDF. Please refer to our response to Reviewer UC6f's Weakness 2 for details of these ablation studies.
>
> [1] J. Ye, Z. Wu, J. Feng, T. Yu, and L. Kong. Compositional exemplars for in-context learning. In ICML 2023.

---

> ### Author Response · Authors · 2024-08-13
> **Follow-Up on Our Rebuttal - Your Feedback is Appreciated**
>
> Dear Reviewer SdUm,
>
> We hope our responses have effectively addressed your concerns and clarified any misunderstandings. We would greatly appreciate it if you could review our rebuttals, especially as the discussion period is set to conclude on **August 13th at 11:59 PM AoE**, which is less than 20 hours away.
>
> Thank you for your time and consideration. We look forward to your feedback.
>
> Best,
> The Authors

---

> > ### Comment · Reviewer_SdUm · 2024-08-14
> > **Reply to authors' rebuttal**
> >
> > I appreciate the authors' clarification and complexcity analysis. Based on these responses, I am now clearer about the contribution and will raise the overall score. Hope the authors can incorporate these into the future version. In addition, I feel it is also interesting to show some qualitative examples reflecting how relevance and diversity combined helps the task.

---

> ### Author Response · Authors · 2024-08-14
> **Official Comment by Authors**
>
> Dear Reviewer SdUm,
>
> Thank you for your thoughtful feedback and for raising the overall score. We are pleased that our clarification and complexity analysis addressed your concerns. We will incorporate these aspects into the future version to further enhance the paper.
>
> We appreciate your suggestion to explore how relevance and diversity enhance ICL. We will delve into these studies in our future work.
>
> Best,
> The Authors

---

### Official Review · Reviewer_3zjF · 2024-07-14

**Soundness:** 2
**Presentation:** 2
**Contribution:** 2
**Rating:** 5
**Confidence:** 4

**Summary:**

This paper presents a framework called Mixture of Demonstrations (MoD) for selecting demonstrations to improve in-context learning (ICL) performance of large language models (LLMs). Their method partitions the demonstration pool into groups, each managed by an expert, and applies an expert-wise training strategy to filter the helpful demonstrations. Their experiments show that MoD on GPT-Neo can outperform other ICL baselines.

**Strengths:**

- The goal of this paper is very clear: the authors aim to optimize the demonstration selection process for ICL.
- The authors propose an MoD framework with expert-wise training strategy, and conduct extensive experiments with 12 NLP tasks, including both classification and generation tasks, to show the effectiveness of their method.
- The authors also conduct ablation studies on different modules, and robustness study on larger LLMs.

**Weaknesses:**

- The computational complexity of the proposed method seems very high. Since in-context learning it self is an inference-based method without training, it would be much more expensive to conduct a training-based method before doing in-context learning on a new task. Moreover, the proposed method needs to do expert-wise training, which further increases its usage of computing resources.
- A related problem is that the authors do not mention which retriever model they use in their methods. Nowadays in-context learning can be used on much larger models to get better performance, such as LLaMA-70B, but it would be very hard for practitioners to train such a large retriever model.
- The authors only conduct their main experiments on a very small model (GPT-Neo with the size of 2.7B). Therefore, it is hard to say whether this method would still be effective when applied to larger and stronger models (e.g., LLaMA3-8B). Thus I feel the experiment is not very convincing to me.

**Questions:**

Please refer to the above section.

**Limitations:**

Yes, the authors address limitations in their paper.

---

> ### Author Rebuttal · Authors · 2024-08-07
>
> > **W1**: The computational complexity of the proposed method seems very high. Since in-context learning itself is an inference-based method without training, it would be much more expensive to conduct a training-based method before doing in-context learning on a new task. Moreover, the proposed method needs to do expert-wise training, which further increases its usage of computing resources.
>
> **RW1**: Thank you for pointing this out. First, we would like to clarify that **we do not need to train the LLM in our MoD framework**. As mentioned in lines 198-200, the retriever model for each expert is based on a BERT-based model, which is significantly smaller in size compared to the LLM. **We only use the LLM's feedback as supervised signals to train the retriever model, without training or fine-tuning the LLM.** Regarding expert-wise training, we acknowledge the additional overhead. However, we argue that this overhead is linearly related to the number of experts, which is typically small. Additionally, the retriever model has a relatively small number of parameters, making the additional overhead manageable. As discussed in the Introduction, the overhead introduced by the Mixture of Experts (MoE) mechanism enhances the selection of high-quality in-context learning examples. Moreover, our MoD method does not introduce additional overhead beyond what is already inherent in the MoE framework.
>
> We also want to highlight that MoD reduces computational complexity in practice. First, MoD requires **fewer examples to achieve better performance than the baseline**, thereby reducing the computational complexity of most LLMs due to the short input length. Second, MoD involves **less computation during the selection process** compared to state-of-the-art baselines. We hereby provide a detailed derivation of our computational costs, in comparison to the most competitive baseline CEIL [1] as follows.
>
>
> **Computational Costs**: CEIL primarily discusses the reduction of complexity in two ways: **the number of demonstrations** and **the inference stage**.
>     Since the attention mechanism in most LLMs has quadratic complexity, fewer demonstrations result in shorter input lengths and reduced computational cost. From Table 1 in the attached PDF, we observe that MoD generally outperforms CEIL using only 4 demonstrations compared to CEIL's 16 demonstrations. This shows that **MoD achieves better performance than CEIL with fewer examples**, resulting in less computation complexity compared to CEIL.
>     As for the inference stage, both MoD and CEIL need to compute the similarity between the query and all $N$ demonstrations, denoted the complexity as $\mathcal{O}(T)$. CEIL uses a KNN retriever to select $n$ candidates $(n<<N)$ to narrow the search space. The complexity of selecting top-$n$ candidates is $\mathcal{O}(N+n\log n)$, where $\mathcal{O}(N)$ is to build a max-heap and $\mathcal{O}(n\log n)$ to extract the top-$n$ elements. Then, CEIL uses a greedy algorithm with Cholesky decomposition to reduce the selection complexity from $\mathcal{O}(nL^4)$ to $\mathcal{O}(nL^2)$, where $L$ is the number of ICL examples. Thus, the total complexity of CEIL at the inference stage is $\mathcal{O}(T+N+n\log n+nL^2)$.
>     In MoD, in the worst case, we select the top $L$ elements in one expert, with a complexity of $\mathcal{O}(N+L\log L)$. Thus, the total complexity of MoD at the inference stage is $\mathcal{O}(T+N+L\log L)$. Given $L<n$, **MoD further reduces complexity compared to CEIL at the inference stage**.
>
> In concrete, we believe that our method maintains its practical applicability and offers significant benefits despite the mentioned overheads.
>
> > **W2**: A related problem is that the authors do not mention which retriever model they use in their methods. Nowadays in-context learning can be used on much larger models to get better performance, such as LLaMA-70B, but it would be very hard for practitioners to train such a large retriever model.
>
> **RW2**: Thank you for mentioning this point. To clarify, **we mentioned the retriever model in Lines 198-200, which is BERT-base-uncased**. Thus, since our retriever is not implemented with a large model such as LLaMA-70B, the computational cost of training our retriever is controllable. We apologize if this was not sufficiently clear, and we will make it more explicit in the revised version.
>
> Specifically, we use the LLM's feedback solely as supervised signals to train the retriever model, without training or fine-tuning the LLM itself. Consequently, when employing much larger LLMs such as GPT-3.5 or LLaMA-70B, **we use them only for inference**. Therefore, our method does not face the challenge of training these larger models.
>
> > **W3**: The authors only conduct their main experiments on a very small model (GPT-Neo with a size of 2.7B). Therefore, it is hard to say whether this method would still be effective when applied to larger and stronger models (e.g., LLaMA3-8B).  Thus I feel the experiment is not very convincing to me.
>
> **RW3**:  Thank you for pointing this out. As clarified in line 244, we chose GPT-Neo in our main experiments to **maintain consistency with previous works and ensure fair comparisons**. Nevertheless, we have also included experiments using supervision signals from LLaMA-7B (which is comparable in size to LLaMA3-8B) to train retriever models. In Table 4, we also report the in-context learning performance on GPT-3.5, which has a significantly larger parameter size. The results still demonstrate the superior performance of our method compared to the baselines, indicating that our method remains effective when applied to larger and stronger models.
>
> We acknowledge the importance of providing more experimental results on larger LLMs for greater comprehensiveness. We will add more relevant content and additional experimental results in the revised version.
>
> [1] Ye et al. Compositional exemplars for in-context learning. ICML 2023.

---

> > ### Comment · Reviewer_3zjF · 2024-08-14
> > **Thanks for your rebuttal**
> >
> > The authors have addressed my concerns and I raised my score accordingly.

---

> ### Author Response · Authors · 2024-08-13
> **Follow-Up on Our Rebuttal - Your Feedback is Appreciated**
>
> Dear Reviewer 3zjF,
>
> We hope that our response can address your concerns and clarify any misunderstandings. We would greatly appreciate it if you could review our responses, as the discussion period will end on **Aug 13 11:59pm AoE** in less than 20 hours.
>
> Thank you very much for your time and consideration. We look forward to hearing from you.
>
> Best,
> The Authors

---

> ### Author Response · Authors · 2024-08-14
> **Thank you so much**
>
> Dear Reviewer 3zjF,
>
> We appreciate your recognition of our work and are pleased that our rebuttal addressed your concern. In the revised version, we will emphasize the relevant information to ensure it is clearer and more accessible to readers.
>
> Best,
> The Authors

---

### Author Rebuttal · Authors · 2024-08-07

Dear Reviewers,

We sincerely thank you for your efforts in reviewing our paper. We hope that our detailed clarifications address your concerns. Below is a summary of common issues raised.

- **Complexity**. We emphasize that **MoD reduces computational complexity in practical applications**. MoD requires fewer examples to achieve better performance than the baseline, while reducing the computational complexity of most LMs due to the quadratic nature of the attention module. Additionally, MoD involves less computation during the selection process compared to state-of-the-art baselines. We use the same hyperparameters and encoder settings as CEIL, and thus the total cost is proportional to the cost of each training sample. Our cost/FLOPs is approximately 3/8 of CEIL. We will include more details and comparisons of computational costs in the revised version.

- **More Ablation Studies**. We have conducted additional ablation studies regarding model structures, hyperparameters, and other factors. **The results are detailed in the attached PDF file**. We appreciate the request for these studies and believe that they will offer valuable insights into our framework.

- **Readability**. We appreciate the suggestions to improve readability. We will provide clearer explanations of formula symbols, specify parameter settings, and include more analysis and discussion in the revised version.

We hope these responses could address your concerns and are happy to answer any further questions during the discussion period.

Sincerely,
The Authors

---

### Author Response · Authors · 2024-08-12
**Looking Forward to Your Reply**

Dear Reviewers,

We have carefully addressed your feedback in our rebuttals and provided detailed responses to each of your comments. We believe these clarifications will help in assessing our work more comprehensively.

We would greatly appreciate it if you could review our rebuttals and provide any further feedback, given that the Author-reviewer discussion will be **closed on Aug 13 11:59pm AoE** in no more than two days. We are willing to answer any further questions.

Thank you for your time and consideration. We look forward to your reply.

Best,
The Authors

---

### Decision · Program_Chairs · 2024-09-25

**Decision:**

Accept (poster)

**Comment:**

This paper introduces the “Mixture of Demonstrations” (MoD) method for LLMs. It is designed to improve in-context learning for LLMs by intelligently managing demonstration selection through a mixture of experts approach. This method is tailored to address the common challenge of large search spaces in demonstration retrieval, which is a significant hurdle in deploying effective in-context learning models.

There is general agreement between the reviewers and myself that the proposed approach’s use of a partitioned demonstration pool and an expert-wise training strategy to reduce complexity and increase the diversity of selected demonstrations is innovative. The authors have also conducted comprehensive experiments across a variety of NLP tasks, demonstrating that MoD outperforms the current SoTA methods like CEIL.

There are, however, several concerns regarding this paper, such as the computational complexity of the proposed method and the absence of experiments on larger models. These concerns were mostly addressed by the authors during the rebuttal phase, including additional results and ablation studies, and a detailed computational cost analysis.

Given the originality of the approach and the depth of experimental validation, I lean towards an acceptance for this paper. I recommend that the authors incorporate the detailed computational cost analyses and the new experimental results into the final manuscript.